# Effects of Nanosilver and Heat Treatment on the Pull-Off Strength of Sealer-Clear Finish in Solid Wood Species

**DOI:** 10.3390/polym14245516

**Published:** 2022-12-16

**Authors:** Hamid R. Taghiyari, Dorina Camelia Ilies, Petar Antov, Grama Vasile, Reza Majidinajafabadi, Seng Hua Lee

**Affiliations:** 1Wood Science and Technology Department, Faculty of Materials Engineering & New Technologies, Shahid Rajaee Teacher Training University, Tehran 16788-15811, Iran; 2Department of Geography, Tourism and Territorial Planning, Faculty of Geography, Tourism and Sport, University of Oradea, 410087 Oradea, Romania; 3Department of Mechanical Wood Technology, Faculty of Forest Industry, University of Forestry, 1797 Sofia, Bulgaria; 4Conservation Department, Moghadam Museum, The University of Tehran, Tehran 1137616687, Iran; 5Department of Wood Industry, Faculty of Applied Sciences, Universiti Teknologi MARA (UiTM) Cawangan Pahang, Bandar Tun Razak 26400, Malaysia; 6Laboratory of Biopolymer and Derivatives, Institute of Tropical Forestry and Forest Product, Universiti Putra Malaysia (UPM), Serdang 43400, Malaysia

**Keywords:** coating, heat treatment, nanotechnology, nanosilver, permeability, porous structure, solid wood, thermal modification

## Abstract

Pull-off strength is an important property of solid wood, influencing the quality of paints and finishes in the modern furniture industry, as well as in historical furniture and for preservation and restoration of heritage objects. The thermal modification and heat treatment of solid wood have been the most used commercial wood modification techniques over the past decades globally. The effects of heat treatment at two mild temperatures (145 and 185 °C) on the pull-off strength of three common solid wood species, i.e., common beech (*Fagus sylvatica* L.), black poplar (*Populus nigra* L.), and silver fir (*Abies alba* Mill.), were studied in the present research work. The specimens were coated with an unpigmented sealer–clear finish based on an organic solvent. The results demonstrated a positive correlation between the density and pull-off strength in the solid wood species. Heat treatment at 145 °C resulted in an increase in the pull-off strength in all three species, due to the formation of new bonds in the cell-wall polymers. Thermal degradation of the polymers at 185 °C weakened the positive effect of the formation of new bonds, resulting in a largely unchanged pull-off strength in comparison with the control specimens. Impregnation with a silver nano-suspension decreased the pull-off strength in beech specimens. It was concluded that density is the decisive factor in determining the pull-off strength, having a significant positive correlation (R-squared value of 0.89). Heat treatment at lower temperatures is recommended, to increase pull-off strength. Higher temperatures can have a decreasing effect on pull-off strength, due to the thermal degradation of cell-wall polymers.

## 1. Introduction

From a biological point of view, wood has a continuous porous structure, because cells have interconnecting systems through which liquids can transfer from one to the other [1,2]. In hardwood species, the main cells are fibers and vessel elements, while in softwoods, tracheids perform both functions [1,2,3]. The cell system and porous structure in turn affect many physical and mechanical properties. The size and diameter of fibers, vessel elements, and pits (that connect adjacent cells) are influential in determining the way fluids pass through wood, and also the way paints and resins can penetrate the wood texture, to strengthen or weaken the anchoring effect. The initial spacing between trees cultivated in a location, intercropping with a variety of indigenous plants, drying procedures to decrease moisture contents in wood [4], growing season, extractive content, and the moisture content and hygroscopicity of wood [5,6] are among the myriad factors that affect the size and dimension of cells in wood species, eventually resulting in a wide variance between solid wood species, and even within the same wood species grown in different localities. Moreover, weathering and environmental impacts on wood can also be of great importance, as they affect many wood properties, including the appearance, service life, performance of different coatings and finishes, and also their comparison [7,8,9]. As such, these external factors also greatly affect the preservation and conservation of heritage objects that should be maintained and preserved for future generations [10,11,12].

Owing to the aforementioned biological and environmental factors, uniformity in axial, radial, or tangential directions is uncommon. Therefore, engineers are constantly researching new methods and techniques for modifying various wood species, in order to improve their physical, mechanical, and aesthetic properties and provide a more homogeneous material for the wood-processing industry. Thermal modification and heat treatment are by far the most commercially advanced methods out of all the wood modification processes that have been studied. Thermal modification of wood has long been recognized as a potentially useful and environmentally-friendly wood protection method, to improve its dimensional stability, increase its bio-deterioration resistance, and enhance its resistance to UV irradiation [13,14,15,16,17,18,19,20].

Although this has negative effects on the strength properties of wood, there are various techniques for mitigating these effects [21,22,23]. For instance, thermal modification under different media (such as steam, water, and oil) has been investigated [22,24], and impregnation of specimens with a sodium borate solution (as an alkali-buttering medium) was also tested, to reduce the severity of the negative impact of thermal modification on the strength of wood specimens [24,25]. Temperatures lower than 130 °Conly result in slight changes in material properties, while temperatures higher than 260 °C result in unacceptable degradation of the substrate [13]. Degradation of hemicelluloses (as the main polymer in the wood cell wall) has a positive correlation with the increase in both the temperature of heat treatment and the duration of heat treatment. Degradation of hemicelluloses eventually reduces the swelling in wood indifferent directions [25,26]. However, structural modifications and chemical changes of lignin have also been suggested to be involved in this process [27,28]. In another theory about the reduction in hygroscopicity of wood, it was proposed that it can attributed to a complementary mechanism to mass loss [5]. The authors suggested that new hydrogen bonds are formed after water molecules are forced out of the polymers of the cell-wall microstructure. Some studies also focused on the effects of heat treatment on properties other than the hygroscopicity (dimensional stability) and mechanical properties. These less studied aspects include surface properties and wood printability [29,30,31,32,33].

Heat treatment at extremely high temperatures has a negative impact on the treated wood’s strength properties. Treatment at a low temperature, on the other hand, may not be sufficient to produce satisfactory results, due to the low thermal conductivity of wood. In this connection, increasing the thermal conductivity of a piece of wood facilitates uniformity of heating of both the inners parts and the surface layers. In fact, a low thermal conductivity delays the temperature of the inner parts from increasing as fast as in the outer layer of a piece of wood; which causes overheating and a consequent unfavorable degradation of cell-wall polymers in the outer layers. As a result, metal nanoparticles with high thermal conductivity coefficients [33,34] were used to accelerate the heat transfer to the inner parts of wood bodies and to provide heat treatment uniformity between the surface layers and the core section [26]. Taghiyari et al. [35] reported that nanosilver impregnation allowed wood to be treated at lower temperatures. As wood impregnated with nanosilver has a better thermal conductivity, heat can move more easily throughout the wood samples. As a result, higher temperatures were not required for beneficial treatment effects, thus negating the negative effects induced by high temperatures. However, both processes (heat treatment and impregnation with various nano-suspensions) change the porous structure of wood species at a microscopic level. These changes have been shown to have a significant impact on the permeability (gas and liquid) of solid woods [36], as well as the penetration of coatings and paints into the porous structure, thereby altering their adhesion strength. Although impregnation with different metal nano-particles significantly affects these two properties in solid wood, as discussed above, namely an increased thermal conductivity (affecting heat treatment results) and an altered porous structure in solid wood (changing the permeability of the impregnated specimens), it should be kept in mind that different metal nano-particles and metal compounds may also form new bonds with the main cell-wall polymers (mainly cellulose, hemicelluloses, and lignin), thus also having a significant impact on the mechanical, physical, and even chemical properties of the wood. In this regard, high adsorption energy values between Ag and Cu particles with cellulose and hemicelluloses revealed the formation of new bonds, resulting in an improvement in the physical and mechanical properties both in solid wood species and wood-based composites [26,35,36,37].

Heat treatment techniques are not considered new methods to improve the dimensional stability and moisture content in solid woods, therefore their effects on the different properties of wood and wood-based composites have already been elaborated, including on the surface properties, coating hardness, artificial weathering, UV resistance, and pull-off adhesion strength of a variety of paints, finishes, and coatings [38,39,40,41,42,43,44,45,46]. In this connection, the adhesion of water-based coatings to different heat-treated wood substrates (maple, beech, and hemlock) was reported to decrease, compared to unheated samples [40]. The decreased strength was attributed to the reduced wettability that occurs in heat-treated wood samples [47,48,49]. Two main chemical alterations occur simultaneously, namely degradation of hemicelluloses and an increase in the cellulose crystallinity of samples. However, plasticization of lignin can also be influential in the reduction of the wettability of wood [50]. A combination of all the above-mentioned factors results in the hydroxyl groups in wood-cell polymers being too far out of reach for the paints and coatings to from a strong bond with the wood substrate.

In terms of the improvement in the thermal conductivity of wood, the authors came across few studies on the effects of enhanced and accelerated heat transfer as a result of impregnation with a silver nano-suspension, particularly its effects on the permeability and pull-off adhesion strength of treated wood. Moreover, as the porous structure and permeability value, and the type of cells that are involved in the fluid transfer process, are quite different in softwoods and hardwoods, wood species of both kinds of wood and highly favored in the industry were chosen. In this connection, beech was chosen as a globally popular and well-known wood species. Poplar is a fast-growing species that has been cultivated in different countries, including Iran. Fir is a softwood species that is exported to many countries with low forest resources, to supply the growing need for wood materials for use in low-cost furniture. Based on these facts and their favorability in the regional market, the above-mentioned wood species were chosen. Moreover, as the longitudinal permeability values of softwoods and hardwoods are substantially different to each other, the pull-off adhesion strength values were measured, to clearly demonstrate the relation of the heat treatment and its effects on the pull-off adhesion in cross-section and permeability.

Based on the above-mentioned short literature review and the different biological and environmental factors that affect the porosity system of wood, as well as the paint pull-off adhesion strength of different wood species as popular materials in both modern and historical objects [51,52,53,54], the present study aimed to investigate and evaluate the effects of heat treatment on the pull-off adhesion strength in three nanosilver-impregnated solid wood species. In this connection, two temperatures were chosen. The first temperature for heat treatment was a popular temperature (185 °C). As the specimens were small in size, it was assumed that temperatures higher than 160 °C might not clearly demonstrate the impact of the heat-transfer being facilitated by impregnation with nanosilver. Therefore, a lower temperature (145 °C) was also added to the experiment, to investigate the probable effects of increased heat-transfer caused by nanosilver impregnation on the overall pull-off adhesion strength and permeability.

## 2. Materials and Methods

### 2.1. Specimen Preparation

Beech (*Fagus orientalis* L.) and poplar (*Populus nigra* L.) are two domestic hardwood species with great industrial popularity. The density and mechanical properties of poplar is not as high as beech wood; however, as a fast-growing species, it is cultivated in many parts of Iran, to satisfy the growing requirements for wood materials. Silver fir (*Abies alba* Mill.) is a softwood species that is grown in many countries and exported worldwide for industrial purposes. In Iran, and neighboring countries as well, this wood species (fir) is imported from northern countries to supply the raw materials for the production of inexpensive furniture. Therefore, these wood species were selected in the present research work. In order to measure the pull-off strength in each wood species, thirty tangential specimens were cut. The dimension of the specimens were250 mm × 150 mm × 15 mm. Once the specimens were cut, they were closely inspected, to reject the ones with defects such as knots, fissures, and checks. The selected specimens were conditioned for eight weeks in room conditions, at a temperature of 25 ± 2 °C and 40 ± 3% relative humidity. In order to simulate the conditions in which painted furniture is actually used, the temperature and humidity were used in accordance to those of normal room conditions in indoor spaces in Tehran. Following the conditioning, the surface was sanded with a 100grit sandpaper, windblown to remove the wood dust, and coated with an unpigmented sealer–clear resin. The resin was an organic solvent finish, produced by Pars-Eshen Co. (Tehran, Iran). The technical specifications of the sealer–clear finish are provided in Table 1. As the binder, nitrocellulose was mixed into the resin. Specimens were coated using a spray in two runs. An interval of 12 h was set between the two runs. The final thickness of the coating was measured to be 110 μm using an ultrasonic coating thickness gauge (DeFlesko PosiTector 200, New York, NY 13669-2205, USA). Five dollies (20 mm in diameter) were stuck onto the painted surface of each specimen using epoxy resin. The moisture content of wood specimens was 8 ± 0.5% at the time of the pull-off tests, in order to avoid thermo-hygromechanical behavior of wood that would have affected the mechanical properties of the specimens (ASTM D4541-02) [55].

### 2.2. Nanosilver Impregnation

A 400 ppm aqueous dispersion of silver nanoparticles was used in the present study. It was made using an electrochemical technique for impregnating specimens [56]. The size range of the silver nanoparticles was 30–80 nm. The formation and size of the silver nano-dispersion was monitored by transmission electron microscopy, for which nanosilver samples were prepared to be checked by drop-coating onto carbon-coated copper grids. The pH of the suspension was measured as 6–7; two kinds of surfactants (anionic and cationic) were used in the suspension as stabilizer; the concentration of the surfactants was two-times that of the nano-silver particles. For impregnating the specimens, the empty-cell process (Rueping method) was used in a pressure vessel under 250 kPa of pressure. The vessel was manufactured by Mehrabadi Machinery Mfg. Co. (Tehran, Iran). The pressure was set at 250 kPa for 30 min. Before and after the impregnation process, all specimens were weighed with a digital scale with 0.01 g precision, and their dimensions were measured using a digital caliper with 0.01 mm precision, to measure the density and the amount of nano-suspension absorption. After the impregnation process, all specimens were collectively conditioned for twelve weeks (temperature of 25 ± 2 °C, and relative humidity of 40 ± 3%). The nanosilver-impregnated specimens were dried to a moisture content of 9 ± 0.5% before being heat-treated.

### 2.3. Heat Treatment Process

Specimens for heat treatment (both at 145 °C and 185 °C) were randomly arranged in a laboratory oven. Heat-treated specimens were marked with HT, and specimens impregnated with nanosilver were coded as NS. Specimens were placed on 3 mm thick wood strips, to prevent direct contact with the metal tray in the laboratory oven and consequent overheating. The heating schedule was nearly the same as in previous studies [26]. They were first heat-treated at 145 °C for twelve hours, all in a single run. Then, all HT-145 and NS-HT-145 specimens were taken out and placed under room conditions (25 ± 2 °C, and 40 ± 3% relative humidity). HT-185 and NS-HT-185 specimens continued to be heat-treated at 185 °C for four extra hours. The coding system used in the present project is summarized in Table 2.

### 2.4. Pull-Off Adhesion Strength Testing

The pull-off strength values in wood species are measured with a test that measured the maximum force that is required to pull a dolly with a specified diameter off from a substrate. In the present research work, pull-off strength was measured in accordance with the ASTM D 4541-02 standard specifications [55]. An automatic PosiTest^®^ pull-off adhesion testing device (Defelsko, Ogdensburg, NY, USA) was used. It contained a self-aligning spherical dolly (adhered to the wood specimens), Type V. The effective surface of the dolly was 20 mm (Figure 1). Based on Equation (1), the maximum tensile pull-off strength (*X*) that any specific paint, coating, or finish can adhere to the substrate could be calculated in mega Pascal (MPa) values. The breaking points were checked, to find out if the failure occurred in the substrate or in the adhesive layer between the substrate and the dolly. In case the failure occurred in the adhesive, the result of the test was eliminated from the authentic results.
(1)$$X=\frac{4\;\;F}{\pi \;\cdot \;d^{2}}(\mathrm{MPa})$$
where *F* is the maximum force at failure point (kg.m.s^−2^) and *d* is the diameter of the dolly (mm) (ASTM D4541-02) [55].

The moisture content of the specimens at the time of the pull-off adhesion tests was 9 ± 0.5%, and the temperature was 25 ± 3 °C.

### 2.5. Gas Permeability Measurement

The history of permeability measurement goes back to the early 18th century when Nollet (physicist, 1700–1770) sealed containers with animal bladders. It was further investigated by Richard Barrer (1910–1996), who developed the Barrer measurement technique, a non-SI unit of gas permeability [49]. This is considered one of the first scientific methods for the measurement of permeation values. Afterwards, many other methods, techniques, and devices were invented and used to measure permeability and diffusion processes in porous media, including solid wood species and wood-composite materials [57,58,59,60]. In the present study, the longitudinal gas permeability of specimens was measured using an apparatus equipped with a seven-level electronic device; the time measurement was carried out with millisecond precision [60]. Falling water was applied to measure and calculate the specific longitudinal gas permeability values. Twenty cylindrical specimens were cut in a longitudinal direction from each wood species. Specimens were 18 mm in diameter and 30 mm in length. All specimens were inspected as free from any check, knot, split, or fungal defects. The side perimeter of each specimen was covered by silicon adhesive, to allow air flow only in the longitudinal direction of the specimens. The cross-sectional ends of all specimens were trimmed using a cutter blade, to ensure that the openings of vessel elements and cells were fully open. For each specimen, the gas permeability value was separately measured. The measurement was carried out at seven different vacuum pressures and in a single run. In every run, the seven-time measurements were registered, to finally calculate the specific permeability values based on seven water columns. The water in the glass tube of the apparatus was at least 15cm above the starting point of the initial (the first step) time-measuring device (Figure 2). The necessary precautions were made to prevent any leakage from the glass tube or connections.

Each specimen was tested three times, to finally calculate the mean permeability value. Then, the mean superficial permeability coefficient was calculated using equations that were first presented by Siau [1,2]. A correction factor was used (based on the viscosity of air) to calculate the specific gas permeability [1,2,3].

### 2.6. Scanning Electron Microscopy (SEM Imaging)

Scanning electron microscope (SEM) imaging of the wood specimens was carried out using a TESCAN-VEGA II LSH apparatus at the thin-film laboratory, FE-SEM lab (Field Emission), School of Electrical & Computer Engineering (The University of Tehran, Tehran, Iran). The apparatus was produced in the Czech Republic (located in 62300 Brno). A field-emission cathode in the electron gun of a scanning electron microscope provided narrower probing beams at both low and high electron energy levels, resulting in an improved spatial resolution, and also minimized sample charging and damage.

### 2.7. Statistical Analysis

One-way analysis of variance (ANOVA) was conducted for statistical analysis. A significant difference at 95% level of confidence was applied, using the SAS software program (version 9.2) (2010). To distinguish the statistical difference between the different treatments, a Duncan multiple range test was carried out. Hierarchical cluster analysis was performed. This analysis clearly demonstrates potential similarities, and/or dissimilarities, between a variety of treatments [62]. The analysis in the present study was performed based on two property values simultaneously, namely unpainted and painted pull-off strength values. SPSS/20 (2010) software was utilized to perform the analyses.

## 3. Results

The results of density measurement indicated that beech had the highest density (0.57 g/cm^3^), followed by fir (0.42 g/cm^3^) and poplar (0.35 g/cm^3^). Weight measurements taken immediately after the impregnation in the pressure vessel demonstrated that the NS-uptake did not correspond to the density of the three wood species. The NS-uptake was measured as 0.38, 0.27, and 0.09 (g/cm^3^) for beech, poplar, and fir wood, respectively. Weight measurements of the specimens before and after the heat treatment demonstrated that the maximum and minimum mass losses were found in the HT185 beech and HT145 poplar specimens, respectively (Table 3). Beech specimens generally showed the highest mass losses in each treatment compared to their poplar and fir counterparts.

The gas permeability measurement in the untreated wood species revealed that the highest values were for the poplar sapwood specimens (3.1 × 10^−13^ m^3^.m^−1^). The mean gas permeability in beech specimens was measured 0.74 (×10^−13^ m^3^.m^−1^), indicating that the gas permeability in poplar was four-times higher compared to that of the beech specimens. The fir wood specimens had the lowest gas permeability, which was measured at 0.003 (×10^−13^ m^3^.m^−1^).

The maximum and minimum pull-off adhesion strength values were found in the unpainted control beech specimens (9.7 MPa) and the painted nanosilver-impregnated poplar specimens heat-treated at 185 °C (2.46 MPa), respectively (Figure 3). In comparison to the other two wood species, the beech specimens had significantly higher pull-off values. It can be seen that heat treatment at 145 °C improved the pull-off strength of beech wood. However, nanosilver impregnation reduced the pull-off strength of beech wood, both painted and unpainted. The pull-off adhesion strength values of the poplar and fir specimens were generally similar, though there were fluctuations in each treatment combination. In every treatment and wood species, the values in the unpainted specimens were higher than those of the painted specimens.

## 4. Discussion

The poplar specimens had the highest gas permeability values, according to the measurements. The high permeability values of poplar wood were attributed to the fact that this wood species has simple perforation plates. Therefore, fluids (in this case, air) could pass through the open simple perforation plates with no physical obstacles. Moreover, the extractives in poplar sapwood were reported to be quite low, resulting in an easier fluid transfer [63]. Unlike poplar, beech wood has both simple and scalariform type perforation plates. The scalariform perforation plates are considered a physical obstacle to fluid transfer [64]. In addition to this, vessel elements in some wood species, such as beech wood, are sometimes partially or wholly blocked by different extractives, settlements, and tyloses [1,2,3,36]. Ultimately, fluids cannot be transferred as easily through beech wood as they can through poplar sapwood, resulting in a lower gas permeability in comparison to the permeability values of poplar. In fir wood, as a softwood species, the whole process of the transfer of fluid through the wood texture is different. Softwood species lack vessel elements that allow the transfer of liquids from leaves to the body and roots of trees, and vice versa. The pits that exist in softwoods’ cell walls act as a bottleneck for the transfer of liquids, resulting in a very low permeability in softwood species [1,2,3]. The dimensions of pits was significantly smaller in comparison to the open poplar vessel’s inner diameter (Figure 4A,B). It was reported that the permeability in hardwood species can be more than 1000-times higher than in softwood species [1,2]. Apparently, the very low NS-uptake in fir wood (0.09 g/cm^3^) can be attributed to its very low permeability.

The pull-off mode of failure in all specimens showed that the failure occurred in the woody substance of the specimens. This demonstrated that the process of sticking the specimens to the dollies was correctly carried out. This also indicated that the measurement of pull-off strength values was significantly influenced by the mechanical strength of the substrate. As such, the pull-off strength values in the beech wood were all significantly higher than those of the poplar and fir wood (Figure 3). This was attributed to the higher density of the beech wood in comparison to the other two wood species. Ahigher woody mass resulted in higher mechanical properties in beech wood [26], including the pull-off strength. In fact, a greater force was needed to pull the dollies off the surface of beech specimens in comparison to the other two species (poplar and fir wood).

In the fir and poplar wood specimens, the pull-off strength values were rather alike. The overall similarity of these two species was attributed to their close densities. In the control (unpainted) and NS-impregnated treatments of fir and poplar, the pull-off strength values tended to be higher in the poplar specimens in comparison to the fir specimens (Figure 3). These higher values were attributed to the higher permeability in poplar, resulting in a better anchoring of the resin between the specimen and the dolly [65,66,67,68,69]. However, in the painted specimens, the similarity between the pull-off values in poplar and fir were more alike. The similarity in the painted treatments was attributed to the fact that the paint used in the present study was a sealer–clear finish. The sealer part blocked the openings of vessels, fibers, and tracheids, ultimately resulting in an impermeable surface to any resin and adhesive. Therefore, the pull-off strength values were fully dependent on the mechanical strength of the substrate, rather than the anchoring and penetration of the resin into the texture of the wood. This resulted in relatively similar pull-off strength values in the different treatments of the poplar and fir specimens. Thus, it is noted that the relationship between the permeability and pull-off strength in solid wood is a new topic, and therefore, further studies must be carried out to clearly evaluate possible correlations between these two properties.

Impregnation of specimens with nanosilver suspension significantly decreased the pull-off strength in all beech treatments, both painted and unpainted (Figure 3). This decrease was attributed to two main factors. The first factor was the increased permeability caused by dissolving some of the extractives, which allowed the fluid (air) to pass more easily through the vessel elements. A similar increase in permeability, as a result of impregnation with liquids and nano-suspensions, was reported previously [36,60]. The increased permeability and increased surface roughness (caused by the impregnation with the aqueous nano-suspension) allowed the adhesive (between the specimen and dolly) to penetrate deeper into the wood texture, making it unable to actively participate in the process of adhering the dolly to the specimen. The second factor that contributed to the decrease in pull-off strength in the NS-impregnated specimens was the microscopic checks that occur in the cell wall during impregnation in a pressure vessel. The formation of these micro cracks and checks was previously reported to cause a reduction in the mechanical properties of some solid wood species [36,69]. The pull-off strength in NS-impregnated specimens was eventually reduced as a result of these two factors, or better put, mechanisms that occurred concurrently. A general trend of a decrease in pull-off strength as a result of the impregnation with nanosilver was also observed in the other two wood species (poplar and fir), although some treatments showed a slight inconsistency in this regard. This inconsistency from the generally decreasing trend was attributed to the great variability in strength in different specimens, which is an inherent property in solid wood species [26,68].

Heat treatment at 145 °C increased the pull-off strength in all three wood species, both in the unimpregnated and nanosilver-impregnated specimens (Figure 3). This increase was attributed to the formation of new bonds in the cell wall polymers as a result of irreversible hydrogen bonding in the course of water movement within the polymers in the cell wall [5,60]. It was reported that the formation of these new bonds resulted in an increase in the other mechanical properties, such as the bending strength and screw-withdrawal strength [5,26]. In terms of the heat treatment at 185 °C, the results showed that the pull-off strength values were not significantly different from the control values in any of the three species and treatments. Moreover, no general increasing or decreasing trend was observed. Previous studies reported a decrease in the pull-off strength as a result of heat treatment in some solid wood species [32,53], though the cited authors further added that the impact of different finishes was decisive in their final conclusions [53], as well as the thickness of the coatings [30]. Moreover, it was reported that heat treatment resulted in a decrease in coating hardness and scratch resistance in some wood species, including limba, iroka, ash, and chestnut [31]. In the present study, heat treatment at 185 °C was influenced by two simultaneous phenomena. The first phenomenon was the formation of new bonds, as discussed earlier in this section (the irreversible hydrogen bonding), having an increasing effect on the pull-off strength. However, there was a second phenomenon involved in the process, namely the condensation of lignin and thermal degradation of different cell wall polymers at 185 °C. The condensation of lignin and degradation of cell wall polymers ultimately resulted in decreased mechanical properties [13,24,68,70,71]. Microscopic imaging illustrated that the cell walls became thinner and cracked due to thermal degradation at 185 °C [13,24,60] (Figure 5). These two processes were simultaneously active in the present research study, resulting in a largely unchanged pull-off strength in the specimens heat-treated at 185 °C.

Collective cluster analysis of the three species was carried out for the six treatments in each wood species, including 18 treatments altogether. The six treatments in each wood species comprised of the control, NS-impregnated, HT-145, NSI-HT-145, HT-185, and NSI-HT-185. The analysis was conducted based on the pull-off strength values of both painted and unpainted specimens. The results of the cluster analysis demonstrated that all beech treatments were closely clustered together, while they were remotely clustered from the other two wood species (Figure 6). This clearly demonstrated that the significantly higher density, as well as its higher mechanical properties, had a significant influence on the overall pull-off strength. Therefore, if a particular industrial application necessitates a high pull-off strength, density should be considered in the decision-making process. The similarity of cluster-grouping for the other twelve treatments of the other two wood species (poplar and fir wood) implied that density can be more influential in the eventual pull-off strength than the wood species. That is, poplar belongs to the hardwood species, while fir belongs to the softwood species, with quite different biological and chemical structures. However, not much difference was observed among the twelve treatments of these two wood species, considering their rather similar pull-off strength values. Nevertheless, it must be noted that this conclusion is applicable when the paint consists of a sealer part. This sealer part blocks the vessels, pits, and openings, making the woody substrate nearly impermeable to the adhesive that sticks the dolly to the substrate. Further studies should be carried out on a wider range of wood species and wood-finish types, to obtain a conclusive outlook on the way different paints react on substrates made of different wood species. Regression analysis of the three wood species and their treatments demonstrated high and statistically significant correlation in pull-off strength values between the painted and unpainted specimens (R-squared value of 0.89). The high and significant coefficient of determination in the three solid wood species was considered to be corroborating evidence that density was the decisive and most influential factor in the pull-off adhesion strength. Based on the cluster analysis, regression analysis, and the pull-off strength values, it can be concluded that density is the decisive factor in the ultimate pull-off strength value of wood intended for particular applications if the paint and finish consist of a sealer part. Although wood species can ultimately influence the pull-off strength results, the great effect of density puts the effect of species into context.

The present study utilized an industrial approach, and therefore an organic solvent was used, to comply with the general demands of the market. However, further studies on finishes with inorganic solvents and lower toxicity to humans should be carried out, to satisfy environmentally-friendly requirements.

## 5. Conclusions

The effects of nanosilver impregnation combined with heat treatment on beech, black poplar, and silver fir wood were investigated in this study. In general, heat treatment at lower temperature (145 °C) resulted in higher pull-off strength in all three wood species, most likely due to the formation of new bonds between the cell wall polymers. In comparison to untreated wood, wood treated at 185 °C showed no significant difference in pull-off strength. The degradation of cell wall polymers at high temperatures may offset the beneficial effects of new cell-wall polymer bonds. It should be noted that nanosilver impregnation did not lead to beneficial results for the pull-off strength of the treated samples. Overall, density is a decisive factor that influenced the pull-off strength of the samples, and this must therefore be taken into account if an industrial application requires a high pull-off strength. Future studies could be dedicated to the investigation of nanosilver and nanowollastonite and evaluation of their potential for improving the pull-off strength of wood used in historical and heritage objects, aiming at the better preservation and conservation of these priceless objects for future generations.

## Figures and Tables

**Figure 1 polymers-14-05516-f001:**
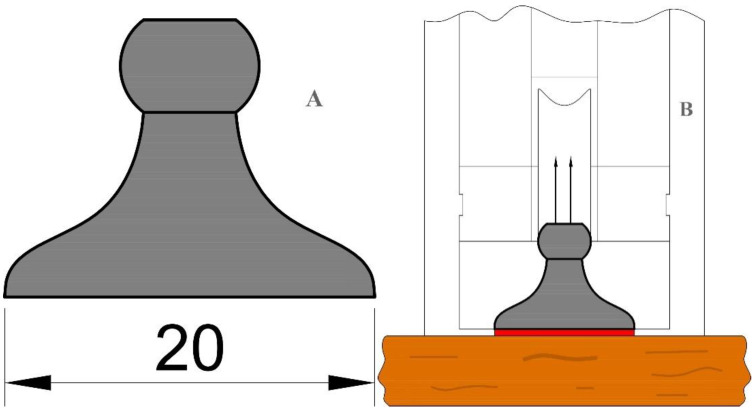
Schematic design of the apparatus to measure pull-off adhesion strength; (**A**) dolly with an effective diameter of 20 mm, (**B**) dolly assembled with adhesive on the substrate.

**Figure 2 polymers-14-05516-f002:**
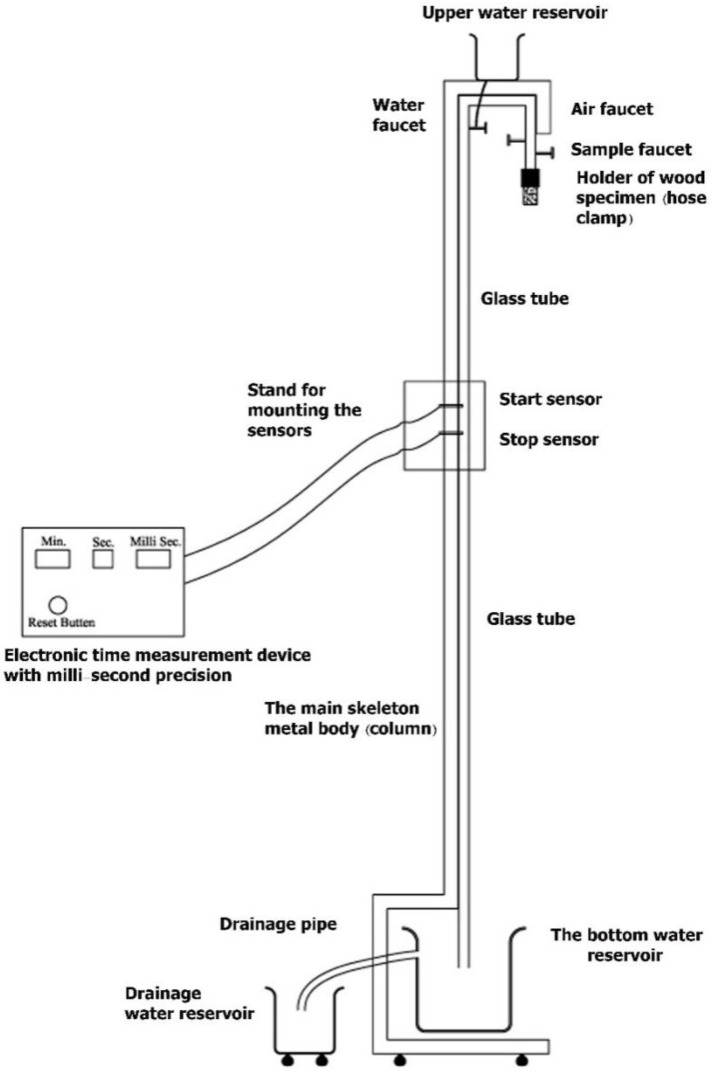
Schematic of the specific gas permeability measurement apparatus, equipped with a single-phase electronic time measurement device with millisecond precision (confirmed by the official certificate No. 47022; issued by The Iranian Research Organization for Scientific and Technology) (USPTO 8079249 B2) [60,61].

**Figure 3 polymers-14-05516-f003:**
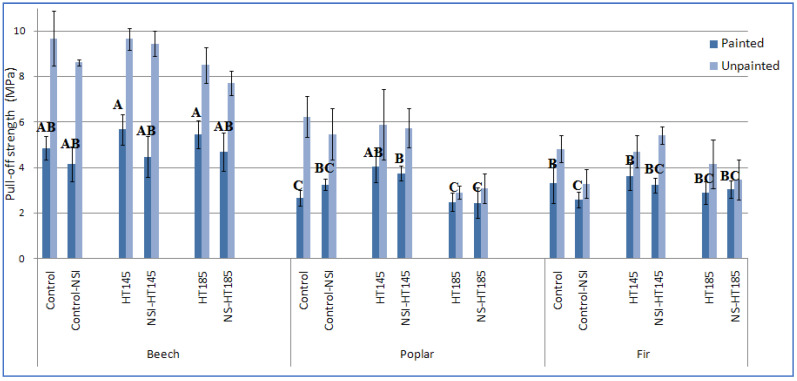
Pull-off adhesion strength (MPa) of the three species (beech, poplar, and fir) painted with sealer–clear finish and unpainted, letters in each column represent Duncan’s multiple range test, at 95% level of confidence (α = 0.05) based on the painted treatments (NSI = nanosilver-impregnated; HT = heat-treated at the specific temperature of either 145 or 185 °C).

**Figure 4 polymers-14-05516-f004:**
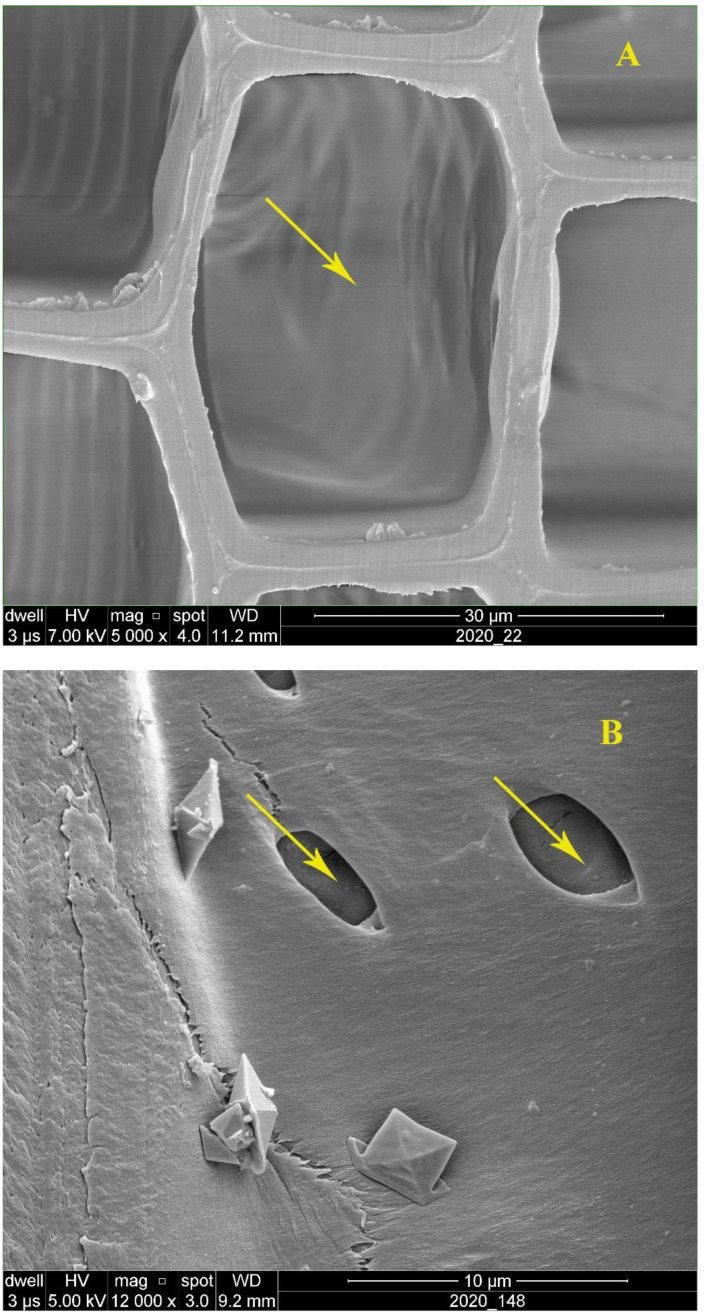
SEM image showing tracheid lumen in the cross-section of fir wood (**A**) and pits in the cell wall in the lateral section (**B**) (↑).

**Figure 5 polymers-14-05516-f005:**
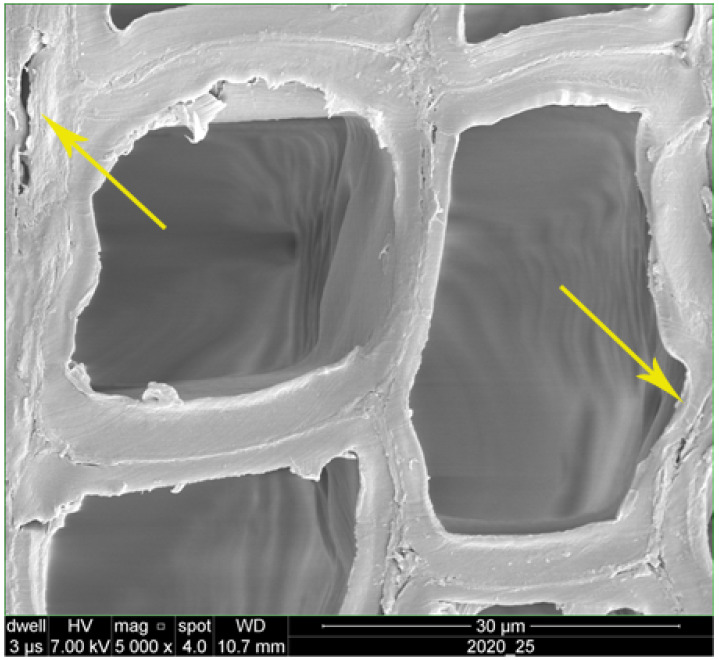
SEM image showing thinned walls and cracks in the cell wall of fir wood (*Abies alba*) heat-treated at 185 °C (↑).

**Figure 6 polymers-14-05516-f006:**
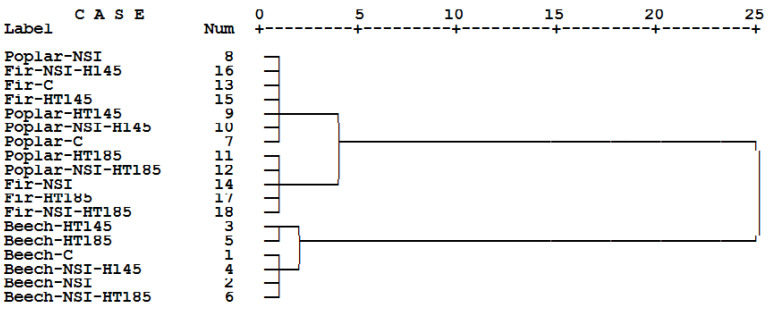
Cluster analysis of the three species (beech, poplar, and fir) based on the pull-off adhesion strengths of painted and unpainted specimens (C = control; HT = heat-treated at the specified temperature; NSI = nanosilver-impregnated specimens).

**Table 1 polymers-14-05516-t001:** Specifications of the sealer–clear finish used in this research work.

Coating Parts	Solids (%)	Viscosity (25 °C) cP	Density (g/cm^3^)	Appearance of the Finish in Liquid Form
Sealer	38 ± 1.5	120 ± 15	0.98	Clear
Clear finish (un-pigmented coating)	39 ± 1.5	80 ± 15	0.98	Clear

**Table 2 polymers-14-05516-t002:** Description of the coding system of different treatments performed in this research work.

Coding	Description of the Treatment
Control	Specimens with no impregnation ormodification
Control-NSI	Specimens impregnated with silver nano-suspension
HT145	Heat-treated specimens at 145 °C
NSI-HT145	Nanosilver-impregnated specimens, heat-treated at 145 °C
HT185	Heat-treated specimens at 185 °C
NSI-HT185	Nanosilver-impregnated specimens, heat-treated at 185 °C

**Table 3 polymers-14-05516-t003:** Mass losses in the three wood species caused by heat treatments at 145 °C and 185 °C.

Heat Treatment ^1^		Beech	Poplar	Fir
HT at 145 °C	HT	10.5 (±0.9) ^2^	6.2 (±0.8)	7.8 (±0.9)
	NSI-HT	10.6 (±0.9)	6.3 (±0.6)	8.1 (±0.6)
HT at 185 °C	HT	11.7 (±0.7)	7.8 (±0.7)	9.2 (±0.7)
	NSI-HT	11.4 (±0.6)	8.3 (±0.5)	9.4 (±0.6)

^1^ NSI = nanosilver-impregnated; HT = heat-treated at the specific temperature of either 145 °C or 185 °C. ^2^ Figures in parenthesis represent standard deviations.

## Data Availability

The data presented in this study are available on request from the corresponding author.

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
