# Peer review of "Effects of Nanosilver and Heat Treatment on the Pull-Off Strength of Sealer-Clear Finish in Solid Wood Species"

_polymers, 2022, doi:10.3390/polym14245516_

Round 1

Reviewer 1 Report

The review of a manuscript „Effects of nanosilver and thermal modification on the pull-off strength of sealer-clear finish in solid wood species” submitted by Taghiyari et al. to the Polymers Journal.

Taking into account the dynamic progress in the growing field on nano-solutions, I believe that the topic is justified. I believe that the submitted work complements the information on the properties of wood impregnated with nano-silver solutions. Moreover, the surface properties and permeability are important features from the application point of view and they were not studied before. I believe that the work has a high level of scientific soundness, the results are complemented by extensive discussion and comparisons with other studies. The conclusions are presented based on the results and form a logical sum. I am sending my suggestions below.

-        Keyword informing about the use of nano-silver should be included.

-        Line 66: It would be beneficial to present those techniques which mitigate the negative effect in short way.

-        Line 84: It would be beneficial to briefly present other effects of impregnation with nanosilver on the properties of wood.

-        Line 95: “…variety of paints, finishes, coatings [28-31]. Maybe it would be good for the introduction to summarize this research briefly.

-        Line 120: Which technique was used to measure the final thickness of the coating layer?

-        Line 123: What is the point of including the reference no. 33 to the standard?

-        Line 134-136: Which pressure was actually applied? 3 bars or 2.5 bars?

-        Line 158: Reference is missing near the standard.

-        Line 251: Based on what this statement on extractives was made?

-        Figure 5. The arrow should be corrected.

Overall I find this research work very interesting and up to date. Looking at the perspectives, all that remains is to wish the authors good luck in their further research and I am waiting for the results concerning historic objects.

Author Response

Reviewer 1

(All modifications in the text are in track/change)

1)

-        Keyword informing about the use of nano-silver should be included.

Reply:

The keyword "Nanosilver" was added in the list.

2)

-        Line 66: It would be beneficial to present those techniques which mitigate the negative effect in short way.

Reply:

Yes, a very good idea. Some points were added in this regard.

3)

-        Line 84: It would be beneficial to briefly present other effects of impregnation with nanosilver on the properties of wood.

Reply:

Different effects of impregnation with nanosilver were summarized and mentioned, including an increased thermal conductivity, an altered porous structure, and formation of new bonds between silver nano-particles and cell-wall polymers.

4)

-        Line 95: “…variety of paints, finishes, coatings [28-31]. Maybe it would be good for the introduction to summarize this research briefly.

Reply:

A short summary of the studies and how they affect pull-off strength in paints and coatings was introduced.

5)

-        Line 120: Which technique was used to measure the final thickness of the coating layer?

Reply:

Yes, indeed. The technique and the apparatus was added in the text.

6)

-        Line 123: What is the point of including the reference no. 33 to the standard?

Reply:

It was explained that thermo-hygromechanical properties in wood can affect the mechanical properties in wood, and therefore all specimens had the same moisture content similar to previous studies.

7)

-        Line 134-136: Which pressure was actually applied? 3 bars or 2.5 bars?

Reply:

The pressure on line 134 was corrected to 2.5 bars.

8)

-        Line 158: Reference is missing near the standard.

Reply:

Reference was added at the end of the sentence.

9)

-        Line 251: Based on what this statement on extractives was made?

Reply:

The relevant reference was added in the text and in the References list.

10)

-        Figure 5. The arrow should be corrected.

Reply:

You are right. The arrows were re-positioned to show the thinned wall and cracks. It seems that the arrows were moved during the uploading of the file.

Reviewer 2 Report

I dont understand wich is the normative that explains the test units dimensions.

Also, when they are stablished in laboratory they use 25ºC and 45% of humidity. These are not the normative conditons: 20ºC and 60%.

Author Response

Reviewer 2

(All modifications in the text are in track/change)

1)

I dont understand which is the normative that explains the test units dimensions.

Reply:

The present study was originally based on comparison of different specimens treated under a variety of conditions (including thermal modification, and impregnation with nanosilver). Different values that were obtained were then compared with each other to find out the effects of each and every treatment on the values that were tested according to the standard procedure. If there would be any more information that is needed to be added in the text, we would highly appreciate if the esteemed reviewer specifies. 

2)

Also, when they are stablished in laboratory they use 25ºC and 45% of humidity. These are not the normative conditons: 20ºC and 60%.

Reply:

The esteemed reviewer is absolutely right. We added explanation in the body of the manuscript that the above mentioned temperature and humidity are those of the normal room conditions in Tehran where painted wood are usually used. Therefore, in order to simulate the normal conditions, we chose to condition the specimens under the same normal temperature and humidity.

Reviewer 3 Report

I do not feel qualified to judge about level of English language, but I understood the ideas of authors, which were written in submission,  well.

Lines 38-39: These sentences are not inventions of authors. The proper citations must be added.

Line 81: Explain, how does thermal conductivity accelerate the heat transfer?

Line 102: The effect of thermal modification on pull of strength was investigated on two different levels of temperature. Is it enough? Is only temperature of modification the factor which characterizes the thermal modification?

Line 124: Please, use only SI units.

Line 194: Permeability coefficient was not firstly mentioned by Siau. 

Line 210: Where are the results of Anova, Duncan test...?

The authors describe the pull off test on cross section as they try to explain results by the permeability in longitudinal direction. Realize, that cross section has minor representation in nowadays furniture industry. Therefore, the submission has serious flaws.

Author Response

Reviewer 3

(All modifications in the text are in track/change)

1)

I do not feel qualified to judge about level of English language, but I understood the ideas of authors, which were written in submission, well.

Reply:

We appreciate the esteemed reviewer's comment.

2)

Lines 38-39: These sentences are not inventions of authors. The proper citations must be added.

Reply:

The relevant citations were added in the body of the manuscript.

3)

Line 81: Explain, how does thermal conductivity accelerate the heat transfer?

Reply:

It was explained that increasing the thermal conductivity of a piece of wood facilitates uniformity of heating at both the inners parts, and the surface layers as well. In fact, low thermal conductivity delays the temperature of the inner parts to be increased as fast as in the outer layer of a piece of wood; this makes over-heating and the consequent unfavorable degradation of cell-wall polymers at the outer layers.

4)

Line 102: The effect of thermal modification on pull of strength was investigated on two different levels of temperature. Is it enough? Is only temperature of modification the factor which characterizes the thermal modification?

Reply:

Yes, a very interesting point. In was added in the text that two temperatures were chosen. The first temperature for thermal modification was a popular temperature (185°C). A lower temperature (145°C) was also added to the experiment to investigate the effects of an increased thermal modification by nanosilver impregnation on the overall pull-off adhesion strength and permeability.

5)

Line 124: Please, use only SI units.

Reply:

It was corrected to Kg.m.s-2.

6)

Line 194: Permeability coefficient was not firstly mentioned by Siau. 

Reply:

A brief history of permeability was added in the text.

7)

Line 210: Where are the results of Anova, Duncan test...?

Reply:

The results of the Duncan Multiple Range Test was added in Figure 3.

8)

The authors describe the pull off test on cross section as they try to explain results by the permeability in longitudinal direction. Realize, that cross section has minor representation in nowadays furniture industry. Therefore, the submission has serious flaws.

Reply:

The esteemed reviewer is right about the minor importance of cross section of solid wood in nowadays furniture industry. However, it was added in the text (the last two paragraphs of the Introduction section) that as the porous structures and permeability values, and the type of cells that are involved in fluids transfer process, are quite different in softwoods and hardwoods, wood species of both kinds of wood with a high favorability for the industry were chosen. In the meantime, as longitudinal permeability values of softwoods and hardwoods are substantially different to each other, pull-off adhesion strength values were measured to clearly demonstrate the relation of thermal modification and its effects between pull-off adhesion in cross section and permeability.  

Reviewer 4 Report

Dear Authors

The topic of the article is interesting and your’s idea to connect thermal modification with nano silver impregnation is very important in my opinion especialy in the topic of wood used to the floor.

Introduction

There is no informations about wood species used to the research. Reviewer doesn’t know what was the idea to choose this three species – poplar, fir and beech

Author doesn’t  present any results for similar research conduct by others authors. Pull-off Strength test are popular for coated wood.

Why author used sealer-clear finish with organic solvent It is not environmental friendly chemicals.

Materials and Methods

Point 2.1 - Specimen Preparation

Why you choose this wood species. Poplar is not popular wood, moreover has low mechanical properties the same as fir wood. Beech is ok.

Why did you choose only two temperatures – 145°C and 185°C. Under the 160°C in wood doesn't make any changes. The changes are observed above 160°C. What about heating time. Why did you decide to choose at 12 hour in 145°C and 4 hour at 185°C. Moreover the softwood is more resistant to the temperature then hardwood. Time is also important factor which influence on the wood properties.

Point 2.3 Thermal Modification Process

In my opinion term – „thermal modification” is used to wood modified in inert gas. In your paper were used typical chamber and as I understood air as a atmosphere of modification In my opinion it was heating wood but not thermally modified wood. Please change it in whole article.

Results

You describe the density result of control wood. What was mass loss of three species of after heating in 145°C and 185°C

Discussion

In this point you showed three photos for SEM technique and only one figures (number 6). In my point of view is too lack. In this paragraph you write a little about pull-off strength. Unfortunately I can't find in this paragraph any results from literature. There is no connection between pull-off test and Gas Permeability Measurement – why you measure this properties. In my opinion you made too less research on the coatings. There is a lot of interesting measurements like thickness of coat, UV resistance, hardness of coated wood, cross cut test and else.

I can't find any statistical analysis in this paragraph.

Author Response

Reviewer 4

(All modifications in the text are in track/change)

1)

The topic of the article is interesting and your’s idea to connect thermal modification with nano silver impregnation is very important in my opinion especialy in the topic of wood used to the floor.

Reply:

We appreciate your kind remarks on our study.

2)

There is no informations about wood species used to the research. Reviewer doesn’t know what was the idea to choose this three species – poplar, fir and beech.

Reply:

The esteemed reviewer is quite right. Explanation on how the selection of wood species made was added in the Introduction section as to the reasons the three wood species were chosen.

3)

Author doesn’t  present any results for similar research conduct by others authors. Pull-off Strength test are popular for coated wood.

Reply:

A very good point; we appreciate the esteemed reviewer's comment on this. We added a couple of previous studies on pull-off strength in wood and wood-based materials.

4)

Why author used sealer-clear finish with organic solvent It is not environmental friendly chemicals.

Reply:

The esteemed reviewer is quite right on the environmental aspects of organic solvents. However, the present study had an industrial approach so that the results can be utilized in the market, and for this kind of popular finish for wood, only organic solvent is used. Still, we added a remark at the end of the Discussion section, indicating that further studies must be carried out to utilize solvents and finishes with lower toxicity for human, and with better environmentally friendly aspects.

5)

Point 2.1 - Specimen Preparation: Why you choose this wood species. Poplar is not popular wood, moreover has low mechanical properties the same as fir wood. Beech is ok.

Reply:

Yes, we agree with the esteemed reviewer that poplar is not very popular; nor is it suitable for furniture industry. But it is cheap. It was explained that poplar and fir are two wood species that are used for the production of inexpensive and low-cost furniture. Poplar is cultivated in many parts of Iran. And Fir is imported from the Northern neighbouring countries. Moreover, because of their low cost and availability, some international researchers have also used them in their studies (Keskin and Tekin 2011; Atar et al. 2015; Ozdemir et al. 2015).

6)

Why did you choose only two temperatures – 145°C and 185°C. Under the 160°C in wood doesn't make any changes. The changes are observed above 160°C. What about heating time. Why did you decide to choose at 12 hour in 145°C and 4 hour at 185°C. Moreover the softwood is more resistant to the temperature then hardwood. Time is also important factor which influence on the wood properties.

Reply:

We quite agree with the comments by the esteemed reviewer. It was explained in the last paragraph of the Introduction section that due to the small sizes of the specimens, it was probable that the impact of facilitated heat-transfer cannot be distinguished. Therefore, a temperature lower than 160°C was also added to the experiment to compare the results with those of the usual thermal modification temperature, that is 185°C. Still, it is explained in the Discussion section that another important point in heat treatment at temperatures lower than 160ºC is the formation of irreversible bonds caused by the movement of water molecules. Therefore, it can be considered that lower temperatures can also be effective, though the molecular structure in wood specimens is not altered to a great extent.

Also, it was explained that the heating schedule was planned similar to some previous studies.

7)

Point 2.3 Thermal Modification Process: In my opinion term – thermal modification” is used to wood modified in inert gas. In your paper were used typical chamber and as I understood air as a atmosphere of modification In my opinion it was heating wood but not thermally modified wood. Please change it in whole article.

Reply:

"Thermal modification" was changed to "heat treatment" all throughout the article.

8)

You describe the density result of control wood. What was mass loss of three species of after heating in 145°C and 185°C.

Reply:

Yes, very good data to be added in the manuscript. Mass losses were calculated based on the weight measurements just before and after the heat treatment. Table 3 was added in the Results section.

9)

In this point you showed three photos for SEM technique and only one figures (number 6). In my point of view is too lack. In this paragraph you write a little about pull-off strength. Unfortunately I can't find in this paragraph any results from literature. There is no connection between pull-off test and Gas Permeability Measurement – why you measure this properties. In my opinion you made too less research on the coatings. There is a lot of interesting measurements like thickness of coat, UV resistance, hardness of coated wood, cross cut test and else.

Reply:

We quite agree with the esteemed reviewer to add more interesting results from other studies. We added a number of different references in the Discussion section.

As to the number of Figures in this section, we might point out that we had to bring the results of the pull-off experiments in the previous section (Fig. 3), because the results had to be first mentioned there. However, we repeatedly referred to the results of the pull-off tests (Fig. 3) in the Discussion section, too. In terms of the permeability measurement and its relation to pull-off strength, we have actually found some deductions here that implicitly indicate relationship between permeability and pull-off in terms of "anchoring". This aspect of the relationship between an increased permeability in wood and an improved anchoring has a novelty and therefore further studies must directly focus on this to come to a concrete conclusion. We added this point in the Discussion section.

10)

I can't find any statistical analysis in this paragraph.

Reply:

The esteemed reviewer is right. Duncan’s Multiple Range test was added to the results of pull-off strength test in Figure 3.

Round 2

Reviewer 3 Report

The 204th-206th lines: Please, use SI units in whole text of the submission.

The 242nd line: N is OK, also does kg∙m∙s-2 (small k)

The 91st – 98th lines: Please, if it is possible, do not clarify. Also, the 388th -391st lines show your poor knowledge about wood.

Author Response

Reviewer 3

(All modifications in the text are in track/change and Yellow highlight)

1)

The 204th-206th lines: Please, use SI units in whole text of the submission.

Reply:

The units “bar” was modified to SI unit kPa.

2)

The 242nd line: N is OK, also does kg∙m∙s-2 (small k)

Reply:

The unit “Kg” was corrected to “kg”.

3)

The 91st – 98th lines: Please, if it is possible, do not clarify. Also, the 388th -391st lines show your poor knowledge about wood.

Reply:

The part of clarification was deleted.

Reviewer 4 Report

Dear Authors

I read carefully your manuscript again. In my opinion now is much better. Of course I find some problems

1) Please add statistical analysis to the table with mass loss informations (table 3 from your answer to my review, table 2 from the manuscript)

2) Please change the hemicellulose to hemicelluloses (it's a group of short degree of polymerization sugars)

3) In Introduction you wrote that poplar and fir are used to produce the low-cost furniture. I agree with this, but in my opinion low cost furniture should be cheap. If furniture factory use to finish the surface some kind of coating the price for this products grow up, unfortunately

Author Response

Reviewer 4

(All modifications in the text are in track/change and Yellow highlight)

1)

1) Please add statistical analysis to the table with mass loss informations (table 3 from your answer to my review, table 2 from the manuscript).

Reply:

The standard deviations of mass losses were added next to the mean values in Table 3.

We believe the esteemed Reviewer meant Table 1 (and not Table 2). The producer of the paint had not provided us with the standard deviation of the Density. And, as we mixed the paint container just before using it, and because it was a liquid, we didn’t measure the density or standard deviation.

2)

2) Please change the hemicellulose to hemicelluloses (it's a group of short degree of polymerization sugars).

Reply:

Yes, thanks. We corrected this point all over the manuscript.

3)

3) In Introduction you wrote that poplar and fir are used to produce the low-cost furniture. I agree with this, but in my opinion low cost furniture should be cheap. If furniture factory use to finish the surface some kind of coating the price for this products grow up, unfortunately.

Reply:

Yes, we quite agree with the esteemed reviewer on this point. However, finishes and coatings are an inseparable part of the furniture in our local market. That is, unpainted furniture is not very popular. Therefore, only the wood and furniture fabric are decisive factors in the final price of furniture; both are used from not very high quality materials (low density wood like poplar, and low quality furniture fabrics).